# A Novel Method of Ionospheric Inversion Based on Horizontal Constraint and Empirical Orthogonal Function

**Debao Wen \*** , **Yinghao Tang and Kangyou Xie**

School of Geography and Remote Sensing, Guangzhou University, Guangzhou 510006, China;
2112001076@e.gzhu.edu.cn (Y.T.); xkkx@e.gzhu.edu.cn (K.X.)
**\*** Correspondence: wdbwhigg@gzhu.edu.cn

**Abstract:** Tomographic inversion of the ionosphere is a rank-deficient problem. To overcome the above problem, an algebraic reconstruction technique (ART) based on adaptive horizontal constraint and empirical orthogonal function (ARTHCEOF) is proposed. The new algorithm avoids the difficulty of vertically constrained matrix construction and resolves the description of the ionospheric vertical structure by using EOF. To confirm the feasibility and validate the ascendancy of the ARTHCEOF, three algorithms are first tested by using the emulated scheme. The test results show that the ARTHCEOF surpasses the ART and the ART based on the horizontal constraint (ARTHC) in both the inversion accuracy and computational efficiency. Finally, the ARTHCEOF method is applied to invert electron density values using the GNSS measurements during different geomagnetic activities. The tomographic images validate that the ARTHCEOF can reflect ionospheric daily changes in the European region. The altitudinal profiles in a fixed location are illustrated according to the inversion results of ARTHCEOF. Compared with the profiles recorded by the ionosonde station, the altitudinal profiles of ARTHCEOF have a good consistency. In the meantime, the VTEC values are inverted using the CIT results. The differential VTEC values are calculated by means of the inverted VTEC values and ionospheric products of CODE. The differential results further identify the dependability of ARTHCEOF.

**Keywords:** ill-posed problem; ionospheric electron density; algebraic reconstruction technique; empirical orthogonal function; computerized ionospheric tomography





## 1. Introduction

The ionosphere is an important component of the atmosphere above the Earth's surface. The aggregation of free electrons impacts the propagation speed of satellite signals [1,2]. This leads to an extension of the transmission time of radio waves, which is referred to as ionospheric delay. For a high-accuracy user, the existence of ionospheric delay error affects the accuracy of satellite positioning [3,4]. To meet the demands of high-accuracy users, it is particularly necessary to detect three-dimensional ionospheric fine structures. The constructions of the global navigation satellite system (GNSS) provide a new ionospheric sounding method. Using the GNSS measurements, ionospheric vertical total electron content (VTEC) values are computed. Although the VTEC images can reflect ionospheric horizontal variations, it neglects ionospheric vertical structures since VTEC values are computed using a thin-layer hypothetical ionospheric model [5–7].

In 1988, the computerized ionospheric tomography (CIT) technique was introduced to probe ionospheric structure by using the measurements of the navy navigation satellite system (NNSS) [8]. Although NNSS-based CIT successfully captured ionospheric vertical structures, it neglected the ionospheric variations in longitudinal direction [9–11]. Using the emulated GNSS data, three-dimensional CIT results were successfully inverted [12]. Then, the GNSS-based CIT technique attracted the attention of some scholars and became the research highlight in the ionospheric field [13–19]. In general, the GNSS-based CIT technique

is a rank-deficient problem due to the inadequacy of GNSS measurements. To surmount the rank deficiency, some algorithms have been proposed [20–26]. The algebraic reconstruction technique (ART) is very popular due to its computational simplicity. However, some pixels do not have any GNSS observation information, which results in the final solutions of these pixels being the same as their priori information after iteration convergence. Some scholars introduced different horizontal constraints to overcome the deficiency of ART to some degree [27–29]. The type of the algorithms is ART based on the different horizontal constraints (ARTHC). However, the studies omit the vertical constraint. This decreases the accuracy of ionospheric inversion. In practice, the vertical constraint matrix is difficult to create due to the complex variation characteristic of altitude. Empirical orthogonal function (EOF) has the advantage of describing the ionospheric vertical structure. Instead of imposing vertical constraint, EOF is used to establish the correlation of adjacent pixels in the vertical direction. The ARTHC based on EOF (ARTHCEOF) is proposed. To examine the ARTHCEOF method, the emulated experiment is made. Finally, the ARTHCEOF method is applied to reconstruct ionospheric electron density (IED) distributions by using the GNSS measurements during different geomagnetic activities. The reconstructed images show the daily variation rules of the IED. The comparisons of ionospheric vertical profiles and the differential VTEC values validate the dependability of the ARTHCEOF.

## 2. Materials and Methods

### 2.1. Basic Principle of CIT

For IED tomographic inversion, the input observation data are the ionospheric slant TEC (STEC) value, which is the IED integral along the propagation paths of the GNSS signal. The formula can be given as:

$$STEC = \int_p N(l)dl \tag{1}$$

where $p$ is GNSS ray path; $N(l)$ represents the IED; $dl$ is an extremely short segment on the ray propagation path. To simplify IED inversion, the reconstructed area is first divided into some small pixels. The IED of each pixel is assumed to be constant in the inverted time period, and then Formula (1) is linearized as:

$$y_i = \sum_{j=1}^{n} A_{ij}x_j \quad i = 1, 2, \cdots, m \tag{2}$$

Taking into account the input STEC noise and discretized error, the matrix expression of Formula (2) is as follows:

$$Y_{m \times 1} = A_{m \times n}X_{n \times 1} + E_{m \times 1} \tag{3}$$

where $n$ stands for the pixel amount; $m$ denotes the amount of input STEC; $Y$ is STEC vector; $A$ is the constructed matrix; $X$ is the column vector of IED; $E$ is the error column vector.

### 2.2. Theory of ARTHCEOF

ART is one of the classical reconstruction algorithms of IED tomographic inversion. In this work, the iterative initial values are given by using the NeQuick model in order to run ART. It is called a round iteration when all available STEC measurements involve an iteration, and then STEC values are computed by using the iterative result of the last round. The difference can be obtained using the input and computed STEC data. ART uses the difference to correct the IED before iterative convergence. The equation of ART can be given as:

$$x_j^{(k+1)} = x_j^{(k)} + \gamma_0 \frac{y_i - a_i x_j^{(k)}}{\|a_i\|^2} a_i^T \tag{4}$$

where $x_j^{(k)}$ is the $k$th iterative result of the $j$th pixel; $a_i$ represents the $i$th row of A; $\gamma_0$ is the relaxation operator affecting the iterative speed and reconstructed quality of ART; $y_i$ indicates the $i$th row of the input STEC vector

In light of Equation (4), ART can modify the IED priori information of each pixel having observation information. In fact, many pixels do not have any GNSS ray traversing them because of the sparse and uneven distribution of ground-based GNSS stations. Their IED priori values cannot be modified in the iterative process. So, the reconstructed accuracy of ART is affected. To cope with the above problem, some scholars impose various constraints in the horizontal direction [27–29]. However, the constraint matrices remain immutably in each round iteration. In addition, vertical constraints are also neglected. To improve the previous work, its elements of the horizontal constraint matrix are anticipated to modify the IED iterative results. The adaptive constrained function is introduced to modify the elements of the horizontal constraint matrix in this work. The horizontal constrained formula can be represented as:

$$f_1 x_1 + \ldots + f_{d-1} x_{d-1} - x_d + f_{d+1} f_{d+1} + \ldots + f_n x_n = 0 \tag{5}$$

The constraint coefficient $f_i$ can be given as:

$$f_j^{(k+1)} = \frac{e^{-\left(b_{cd}^{(k)}\right)^2 / 2\sigma^2}}{\sum\limits_{c=1}^{ne} \sum\limits_{d=1}^{nn} e^{-\left(b_{cd}^{(k)}\right)^2 / 2\sigma^2}} \tag{6}$$

$$b_{cd}^{(k)} = D_{cd} \times \frac{x_d^{(k)}}{x_c^{(k)}} \tag{7}$$

where $ne$ is the pixel amount in the longitudinal direction; $nn$ represents the pixel amount in the latitudinal direction; $D_{cd}$ is the space of the center of two pixels; $\sigma$ is the smoothing operator.

In light of Formula (6), the constraint matrix in the horizontal direction is written as:

$$F = diag\left(F_1, F_2, \ldots, F_g\right) \tag{8}$$

$F_g$ represents the constrained matrix of the $g$th layer. The constrained formula is as follows:

$$FX = 0 \tag{9}$$

Combining Formula (3) with Formula (9), a new formula can be obtained.

$$\begin{bmatrix} Y \\ 0 \end{bmatrix} = \begin{bmatrix} A \\ F \end{bmatrix} X + E \tag{10}$$

The matrix form of Formula (10) is given as:

$$B_{(m+h) \times n} X_{n \times 1} + E_{(m+h) \times 1} = L_{(m+h) \times 1} \tag{11}$$

where $h$ is the number of constrained equations.

Considering the stratification and gradient features of the ionosphere in altitude, the construction of the vertical constraint matrix is difficult. So, EOF is introduced to describe the vertical variation of the ionosphere. Substituting the orthogonal matrix composed of EOF vertical basis functions into Equation (11), the following equation can be obtained.

$$B_{(m+h) \times n} \cdot U_{n \times p} \cdot U_{n \times p}^T \cdot X_{n \times 1} + E_{(m+h) \times 1} = L_{(m+h) \times 1} \tag{12}$$

$p$ is related to the number of horizontal pixels and EOF. $U$ is a matrix composed of EOF.

In general, the first three orders of EOF can basically describe 95% ionospheric profiles. Figure 1 shows the first three orders of EOF in European regions. From Figure 1, it can be seen that the first three orders of EOF include 79.88 percent, 10.08 percent, and 5.47 percent information on the ionospheric vertical profile, respectively.

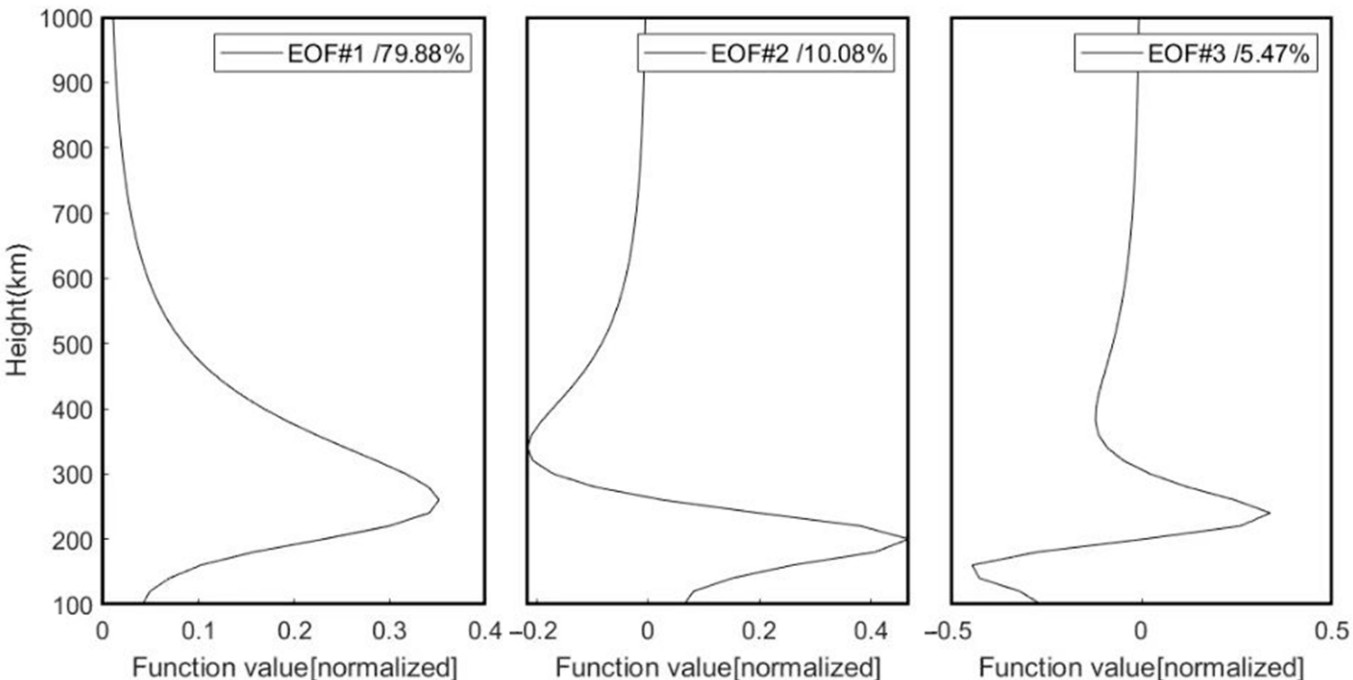

**Figure 1.** Diagrammatic sketch of the first three orders of EOF describing ionospheric information in altitudinal direction.

### 3. Results

*3.1. Simulated Test of the New CIT Algorithm*

To verify the practicability and advantages of ARTHCEOF, numerical examinations of three CIT algorithms are first carried out. The examining results of three algorithms are compared with the emulated IED truth values. In the emulation scheme, the latitudinal range is 36°–60°N and the longitudinal range is 0°–32°E. The scope changes from 100 km to 1000 km in height. Nesterov and Kunitsyn [30] validated that the spatial resolution can be selected as 2°–4° when GNSS stations are dense. To construct the coefficient matrix of CIT, the spatial coordinates of 183 European GNSS stations are used. The geographic locations of the GNSS and ionsonde stations are illustrated in Figure 2, which shows that the selected European GNSS stations are dense. So, the longitudinal step is 4°, and the latitudinal step is 2°. In the altitudinal direction, the interval is 20 km.

The geomagnetic activity is calm at the time period of 13:30–14:00UT on 20 August 2018. The GNSS data of the above time period are selected to examine three algorithms. The IED background values are necessary to run iterative CIT algorithms. In this work, the IED background values are obtained from the NeQuick model. To distinguish the IED background value, the IED true values are given by IRI empirical model. Figure 3 illustrates the images of the IED background information in different transverse sections, and it shows that the imaging distortion is serious.

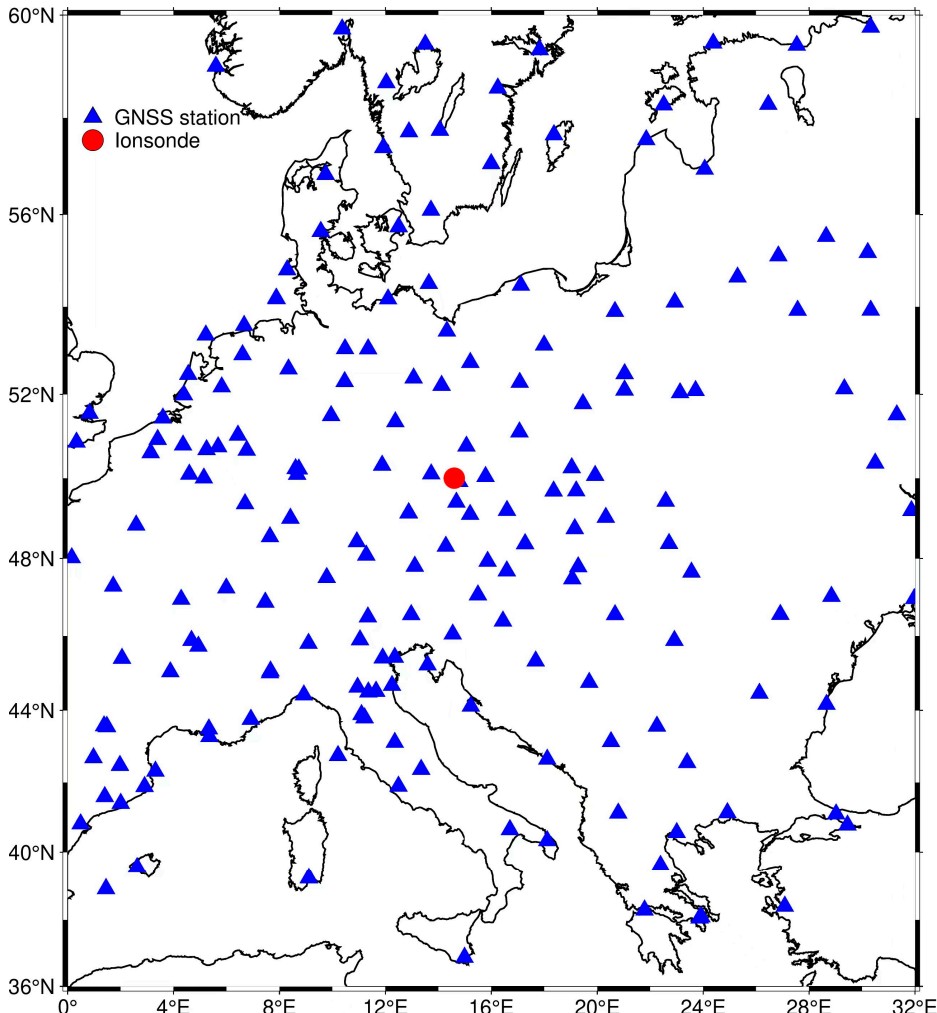

**Figure 2.** The distribution of the 183 European GNSS stations.

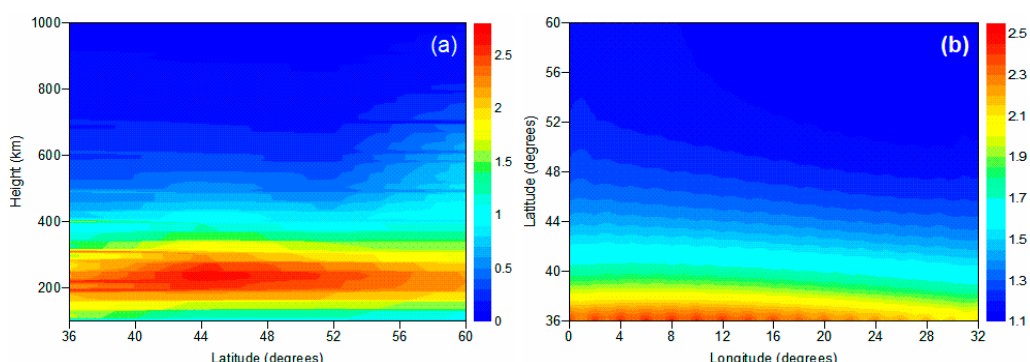

**Figure 3.** Images of IED background information at 14:00UT at 14°E and 305 km. (**a**) IED image along 14°E; (**b**) IED image at 350 km. The IED unit is $10^{11}$ el/m$^3$.

Figures 4 and 5 compare the inversion results of three tomographic algorithms with those reconstructed by the IRI 2016 model in different cross-sections. The comparisons confirm that the inversion results of the ARTHCEOF method is more consistent with the IED true values than those of the ART and ARTHC, and the image distortions of the ART and ARTHC are more severe than those of the ARTHCEOF. The reason is that the ART and ARTHC cannot accurately describe the vertical variation of the ionospheric structure. The inversion results reveal that the IED values gradually decrease with the increasing latitude and longitude at 14:00UT on 20 August 2018.

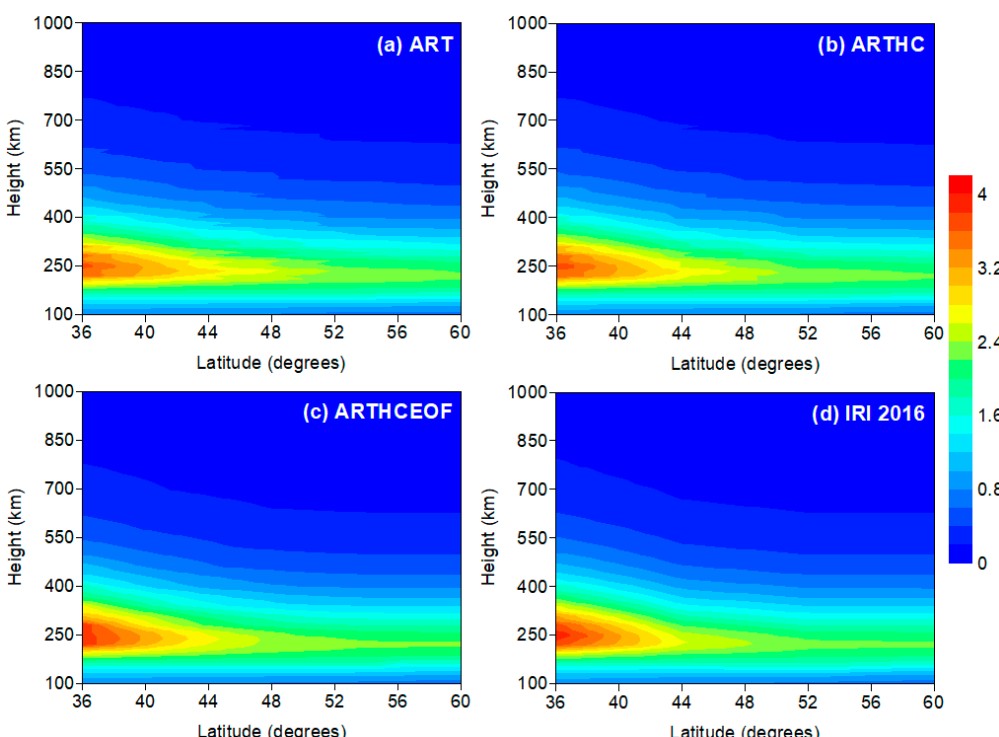

**Figure 4.** IED inversion results of three CIT algorithms and IED true values along 14°E. (**a**) ART; (**b**) ARTHC; (**c**) ARTHCEOF; (**d**) IRI 2016 model. The IED unit is $10^{11}$ el/m$^3$.

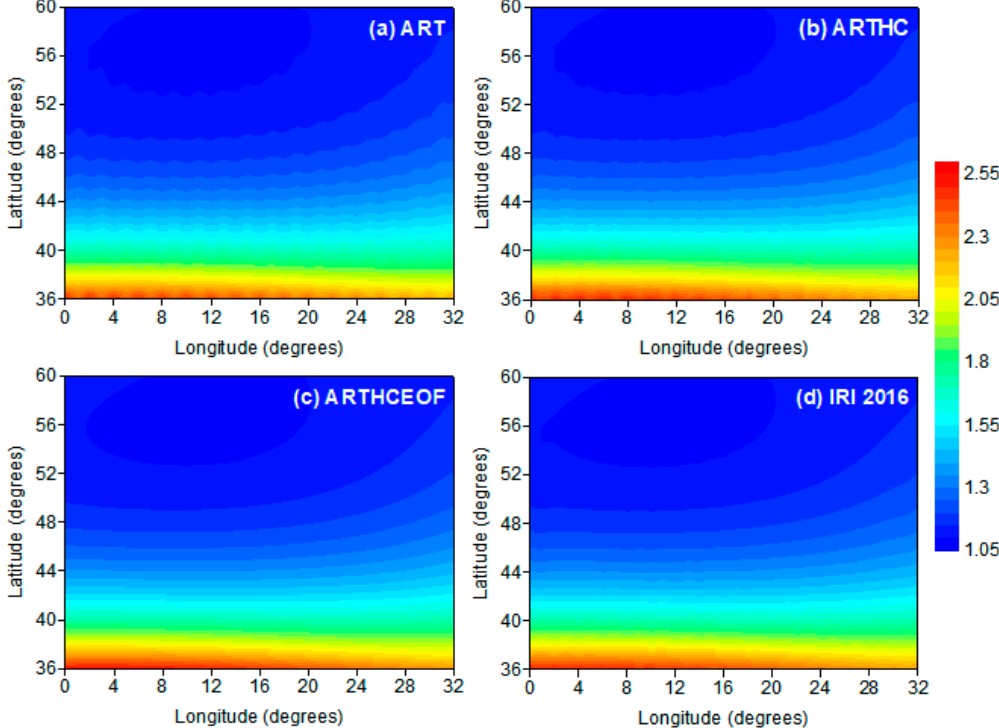

**Figure 5.** IED inversion results of three CIT algorithms and IED true values at 350 km. (**a**) ART; (**b**) ARTHC; (**c**) ARTHCEOF; (**d**) IRI 2016. The IED unit is $10^{11}$ el/m$^3$.

To evaluate the reconstructed accuracy, the selection of evaluating indicators is very important. Three evaluating indicators are introduced in this work. One is the IED

difference (IEDD), another is the mean absolute error (MAE), and the last is the root mean square error (RMSE). Three indicators are calculated using the following formulas:

$$\text{IEDD} = x_j^{tomo} - x_j^{true} \tag{13}$$

$$\text{MAE} = \sum_{j=1}^{n} \left| x_j^{tomo} - x_j^{true} \right| \bigg/ n \tag{14}$$

$$\text{RMSE} = \sqrt{ \sum_{j=1}^{n} \left( x_j^{tomo} - x_j^{true} \right)^2 \bigg/ n } \tag{15}$$

where $x_j^{true}$ is the emulated IED value of the $j$th pixel, and $x_j^{tomo}$ is the CIT result of the $j$th pixel.

The IEDD of three CIT methods is first calculated. Figure 6 counts the number of pixels in each error histogram. Figure 6c reveals that the amount of pixels with small IEDD is significantly larger than those in Figures 6a and 6b. Table 1 shows the error indicators and iterative efficiency of three CIT algorithms after iteration convergence. The census indicates that the inversion accuracy and efficiency of the ARTHCEOF are higher than those of the ART and ARTHC. The emulation results validate that the reconstructed accuracy can be improved by introducing EOF to ARTHC.

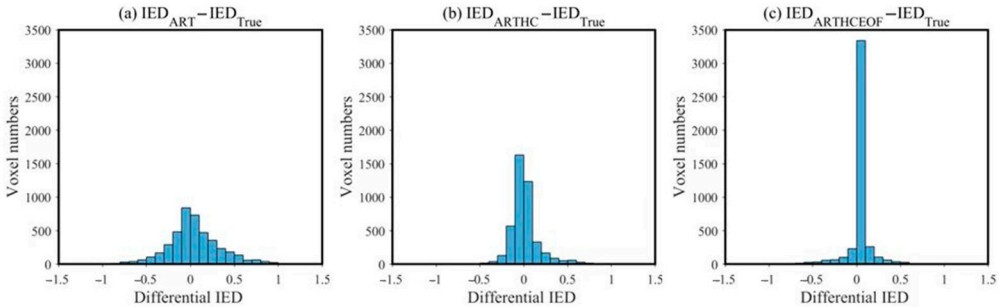

**Figure 6.** Histogram of the reconstructed error statistics. The unit of differential IED is $10^{11}$ el/m$^3$.

**Table 1.** Census of error indicators and iterative efficiency of three CIT algorithms. The IED unit is $10^{11}$ el/m$^3$.

| Methods | ART | ARTHC | ARTHCEOF |
|---|---|---|---|
| Maximum absolute IEDD | 1.48 | 1.28 | 0.56 |
| MAE | 0.23 | 0.11 | 0.06 |
| RMSE | 0.33 | 0.18 | 0.14 |
| Iterative round numbers | 23 | 16 | 10 |

### 3.2. IED Reconstruction during Geomagnetic Quiet Day

To examine the applicability of the ARTHCEOF method, the measured data of 183 GNSS stations shown in Figure 2 are used to compute the input STEC data in the European area. The geomagnetic activity is calm on 24 August 2018, so it is chosen to examine the ARTHCEOF. Carrier phase smoothing pseudorange is introduced to compute the input STEC data. The satellite DCB is corrected through the products released by the Chinese Academy of Science, and the computed formula of the receiver's DCB is as follows:

$$DCB_r = \sum_{i=1}^{m} STEC_i \cdot MF(z) - VTEC_{CODE} \bigg/ m \tag{16}$$

where $VTEC_{CODE}$ is the ionospheric VTEC at the puncture point obtained by bilinear interpolation; $MF(z)$ is the projection function, which can be represented as:

$$MF(z) = \cos\left(\sin^{-1}\left(\frac{R}{R+H}\sin\alpha z\right)\right) \quad (17)$$

where $z$ is the satellite elevation angle; H is the thin-layer height; R is Earth radius; $\alpha$ is the scale factor. To update the accuracy of input STEC measurements, the STEC values of the plasmasphere are removed using the IRI-plas model in this work.

Figure 7 illustrates three-dimensional IED variations in the European area on 24 August 2018. The tomographic results of the ARTHCEOF reveal that European IED values gradually increase as time goes on between 2:00 and 10:00UT, and then the IED begins to decrease. However, an ionospheric anomaly phenomenon occurs at 18:00UT, and the IED values reach the maximum of the day, the phenomenon is consistent with the release's products of CODE. Figure 7 also reveals that the IED values decrease with an increasing latitude. The IED distributions in the east are generally greater than those in the west.

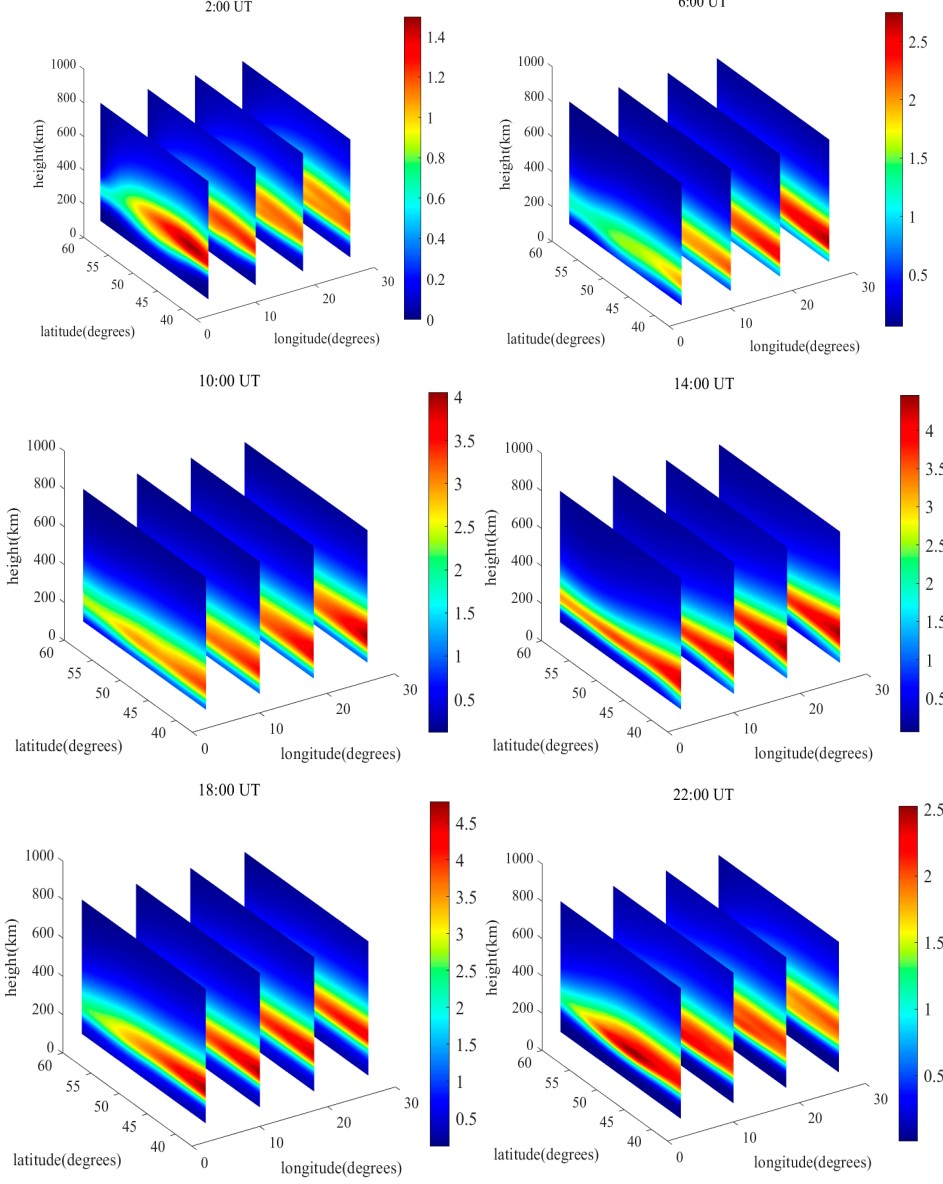

**Figure 7.** Three-dimensional IED distribution images during geomagnetic quiet day. The IED unit is $10^{11}$ el/m$^3$.

To confirm the dependability of the inversion results in Figure 7, the vertical profiles of the CIT are compared with those obtained from the PQ052 ionsonde station, which is located at the 50°N and 14.6°E. The compared images are shown in Figure 8. The comparisons show that the vertical profiles of ARTHCEOF coincide with those given by ionsonde data. It demonstrates that the tomographic results are dependable to some extent. However, the above comparisons are made in a fixed geographical location, and they cannot reflect the dependability of all inversion results. To comprehensively evaluate the dependability of ARTHCEOF, VTEC values are computed using the IED inversion results. By means of the computed VTEC values and the VTEC products of CODE, the differences can be obtained. Figure 9 illustrates the differential VTEC images in six time periods. It shows that the maximum difference is 2.5 TECU, and the minimum difference is −1.5 TECU. The statistics prove that the inverse VTEC values of ARTHCEOF are consistent with those released by CODE. Statistical results of differential VTEC thoroughly validate the dependability of ARTHCEOF.

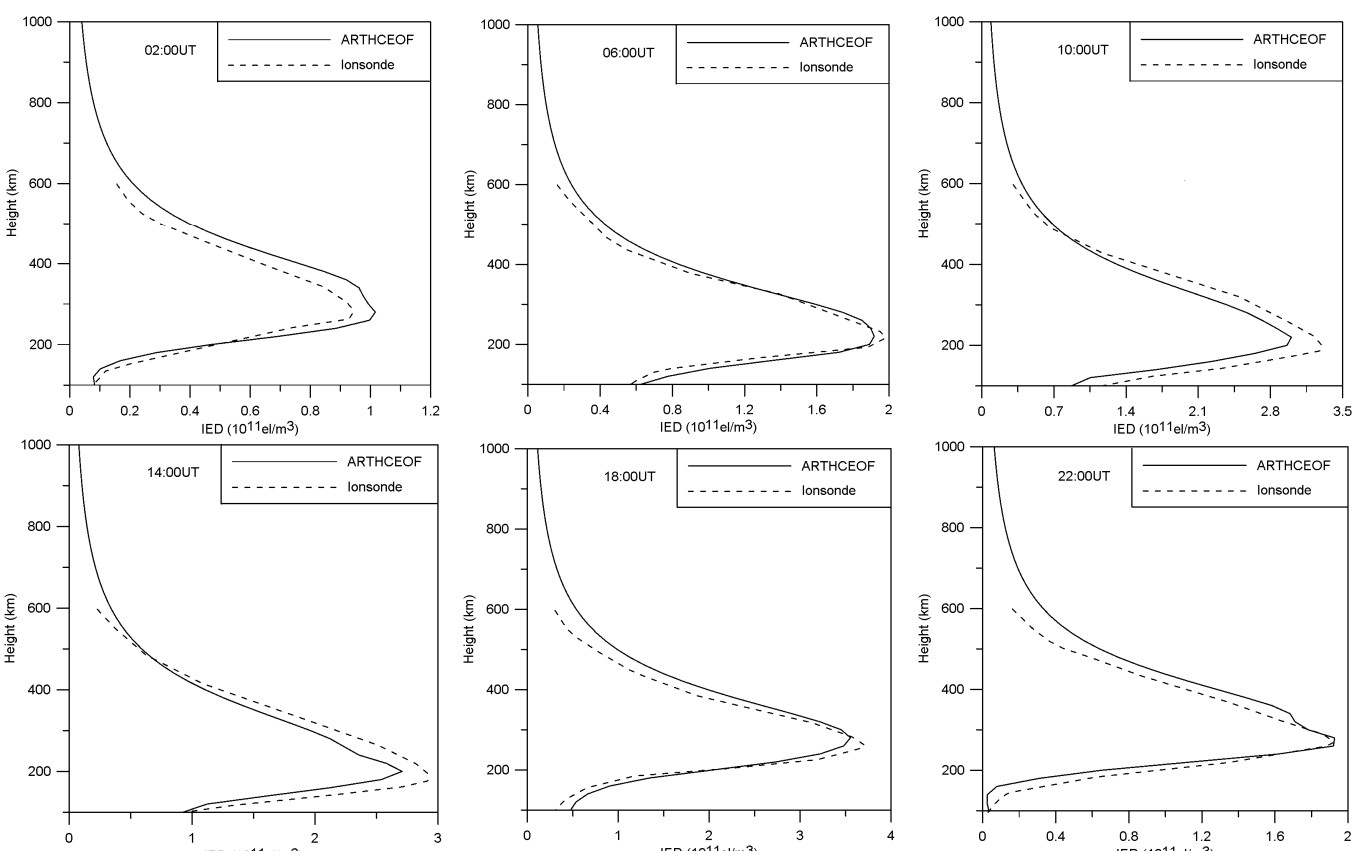

**Figure 8.** Comparisons of ionospheric vertical profiles with those of ionosonde data.

### 3.3. IED Reconstruction during Geomagnetic Storm Day

Due to the influence of the solar coronal mass ejection, a strong geomagnetic storm erupted on 26 August 2018. Figure 10 shows that the Dst index peak reaches −174nT, and the Kp index reaches 8. So, the GNSS data of this day are selected to further examine the characteristics of the ARTHCEOF.

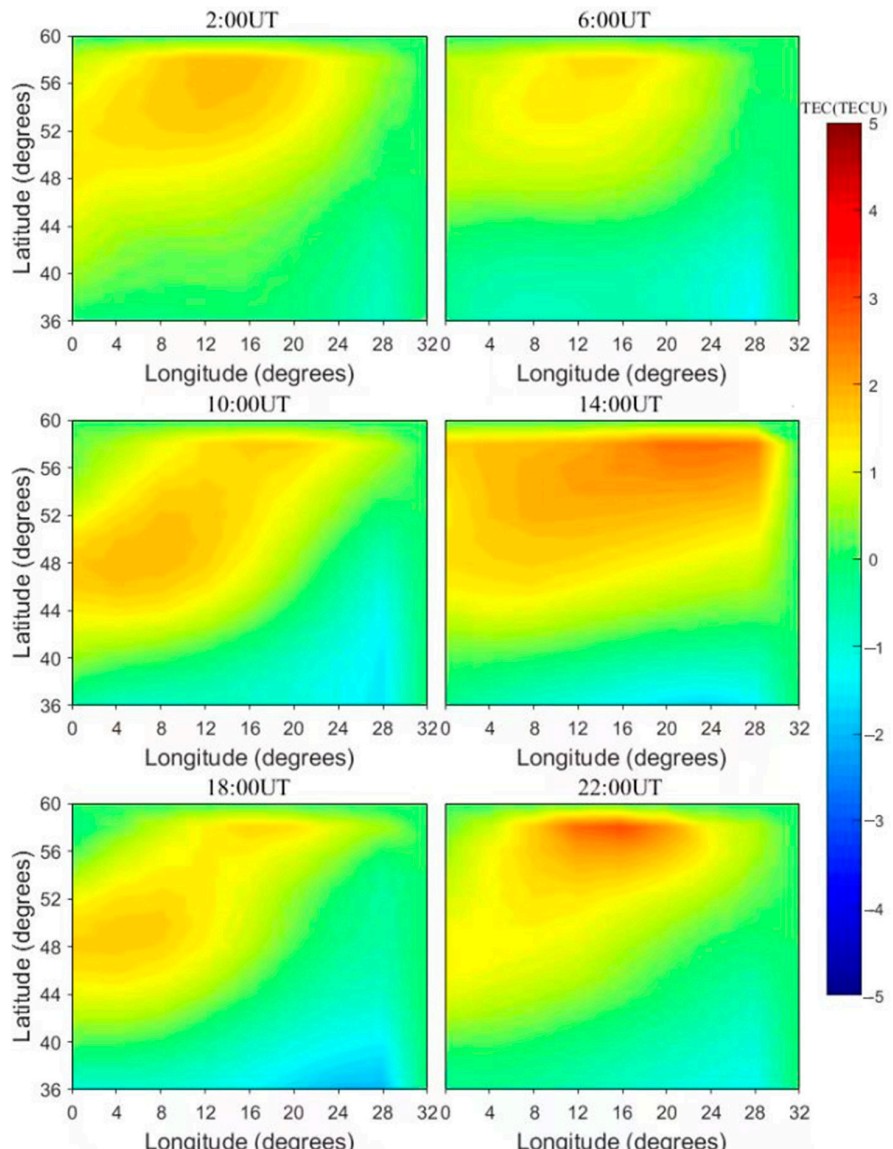

**Figure 9.** Differential VTEC images between the inverted VTEC and the products of CODE during geomagnetic quiet day. The unit of TEC is TECU.

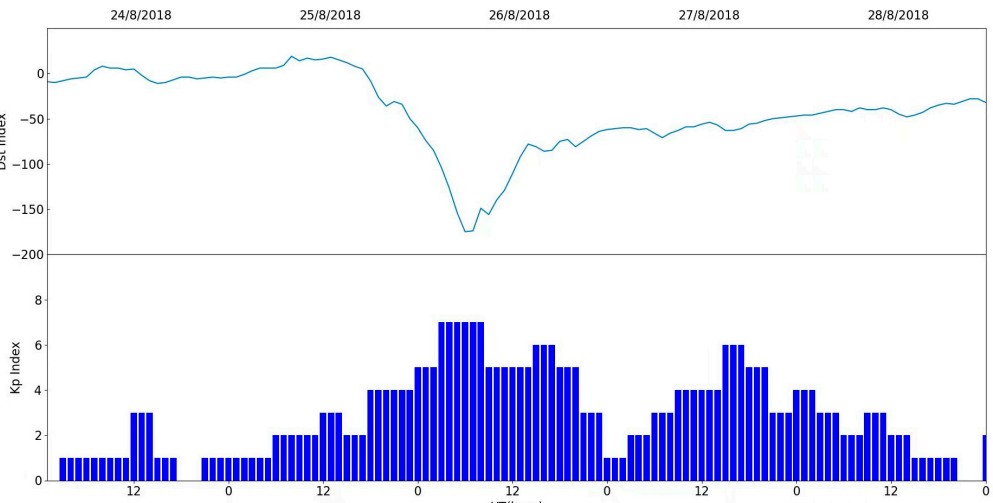

**Figure 10.** Variations of the Dst and Kp indexes during 24–28 August 2018.

Figure 11 illustrates the two-dimensional images at 14:00UT on 25–28 August 2018. The fixed longitudinal chain is 14°E. Figure 11 displays that the IED values decrease at 14:00UT on 26 August 2018. This suggests that a negative phase storm occurs, and then the electron densities begin to augment in the recovery phase on 27 August 2018. However, the IED cannot recover to the level before the geomagnetic storm occurs. Then, the IED values return to the level of the initial phase on 28 August 2018.

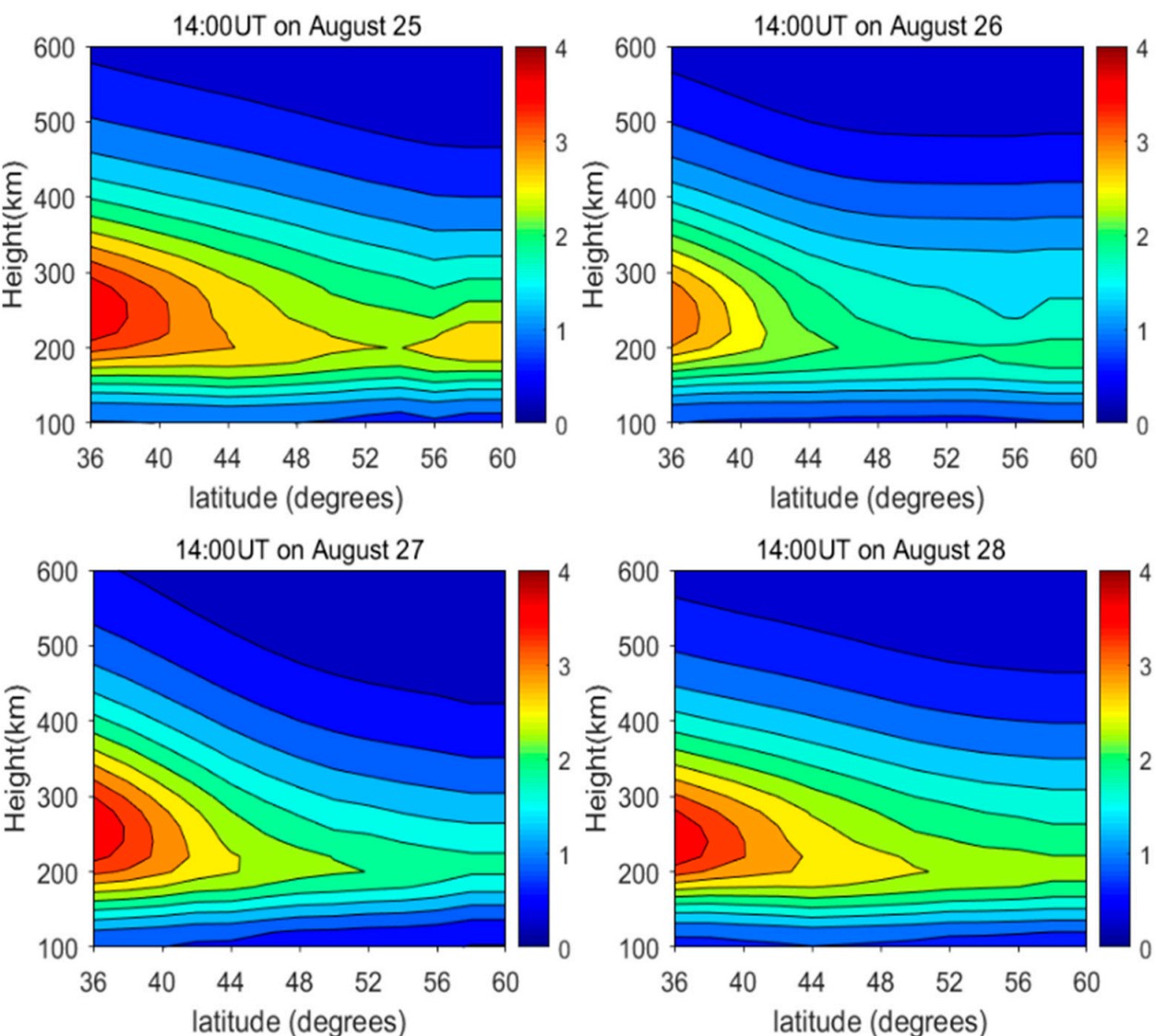

**Figure 11.** Two-dimensional images along 14°E at 14:00 on 25–28 August 2018.

Figure 12 shows the ionospheric daily variations on 26 August 2018. It reveals that the IED qualities augment between 2:00UT and 10:00UT, and then the IED qualities decrease as time goes on. However, the ionospheric variation exhibits abnormal phenomena at 18:00UT. In the meanwhile, it follows that the peak height is 350 km at 2:00UT, and then the peak heights decrease to 250 km between 6:00UT and 14:00UT. Subsequently, the peak height increases to 280 km at 18:00UT. At 22:00UT, the peak height recovered to 350 km.

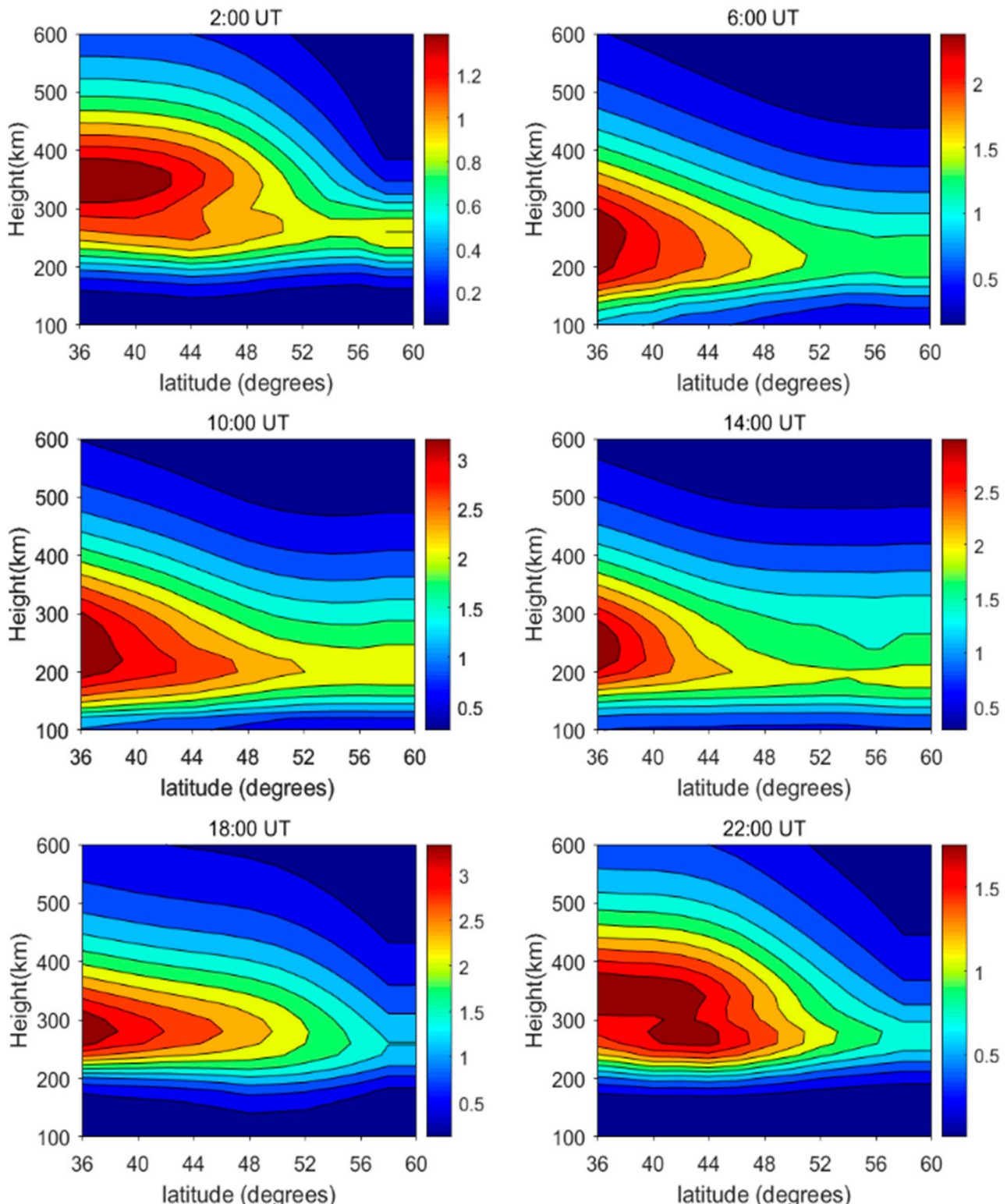

**Figure 12.** IED daily variations along 14°E on 26 August 2018. The IED unit is $10^{11}$ el/m$^3$.

Ionospheric VTEC values are inverted using the tomographic results of ARTHCEOF, and then differential VTECs are calculated by means of the computed VTEC values and VTEC products of CODE are calculated. Figure 13 shows the differential VTEC values of the same time periods in Figure 12. From Figure 13, it follows that the maximum absolute IEDD is less than 2 TECU. This validates the accuracy of the ARTHCEOF methods is comparable

to the accuracy of CODE. The results further verify that the tomographic results of the ARTHCEOF are reliable.

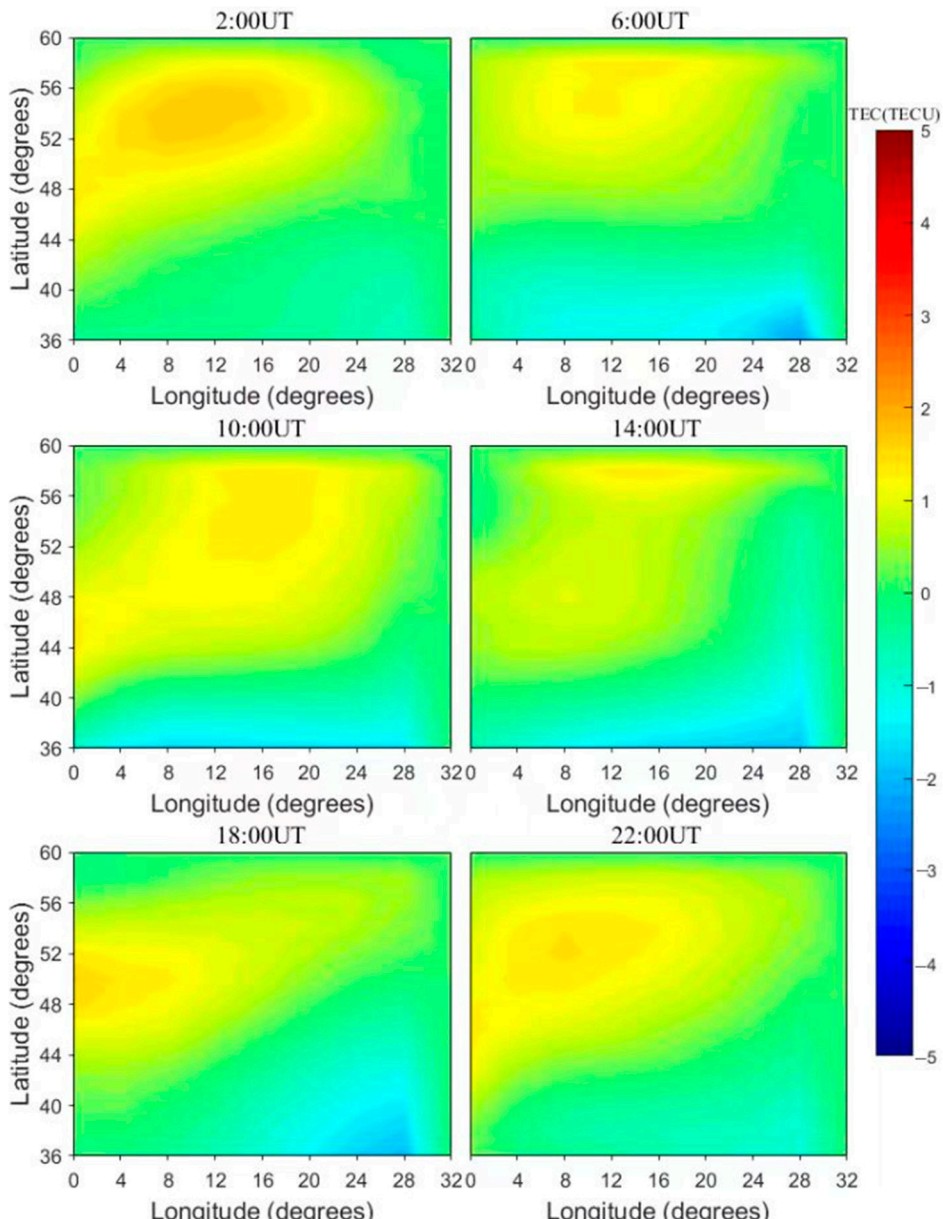

**Figure 13.** Differential VTEC images between the inverted VTEC and the products of CODE during geomagnetic storm day. The unit of TEC is TECU.

## 4. Conclusions

The ARTHCEOF is proposed to reconstruct three-dimensional ionospheric images by using GNSS measurements in the European region. The new algorithm employs EOF to describe the ionospheric vertical structure instead of vertical constraints. It copes with the difficulty of constructing the vertical constraint matrix of the CIT system. The ART, the ARTHC, and the ARTHCEOF are tested using the designed simulation scheme. The numerical scheme demonstrates that the inverted efficiency and accuracy of ARTHCEOF are augmented. Three-dimensional IED images are successfully reconstructed. The CIT results of the ARTHCEOF show the ionospheric daily variation anomaly phenomenon. Finally, an ionospheric storm is used to further test the new algorithm. Compared with the CIT results on 25 August 2018, the ionospheric negative phase storm arises on 26 August 2018. In the meantime, the VTEC are inverted by using the CIT results during

the geomagnetic quiet and disturbed days. The differential VTEC results reveal that the maximum values are smaller than 3TECU and 2TECU during geomagnetic quiet and disturbed days, respectively. The differential VTEC results validate the reliability of the ARTHCEOF method.

Although the ARTHCEOF can successfully capture three-dimensional ionospheric structure, the new algorithm has not been tested in other ionospheric events such as seismicity, solar flare and traveling ionospheric disturbance. In future, the ARTHCEOF will be extended to these fields.

**Author Contributions:** Methodology, D.W.; software, Y.T. and K.X.; validation, Y.T.; formal analysis, D.W.; writing—original draft preparation, D.W.; writing—review and editing, K.X.; funding acquisition, D.W. All authors have read and agreed to the published version of the manuscript.

**Funding:** This study was funded by the National Natural Science Foundation of China, grant number 42070430, the Natural Science Foundation of Guangdong Province of China, grant number 2022A1515011039. Guangzhou City School Joint Project, grant number 202201020136.

**Data Availability Statement:** The readers can obtain the data from the corresponding author.

**Acknowledgments:** Thanks World Data Center and EUREF Permanent GNSS Network for providing ionsonde data and the GNSS data.

**Conflicts of Interest:** The authors declare no conflict of interest.

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
