# Peer review of "A Novel Method of Ionospheric Inversion Based on Horizontal Constraint and Empirical Orthogonal Function"

_remotesensing, doi:10.3390/rs15123124_

Round 1

Reviewer 1 Report

Comments on remotesensing-2447827

The presented work is interesting as a methodological development in obtaining accurate and correct solution. A new approach is proposed to resolve the rank-deficient problem of GNSS-based ionospheric tomography. Three-dimensional electron density distribution is reconstructed using STEC measurements on a certain plane above the Earth's surface. The stability and correctness of the solution is studied for the STEC data synthesized in the IRI-2016 ionosphere model and the results of applying the proposed method to the actual results of GNSS measurements in calm and disturbed ionospheric conditions are presented. Ionospheric vertical profiles of six time-periods are partially compared with those recorded by ionosonde. The comparisons confirm the reliability of the new algorithm. In the meanwhile, the differential STEC further verifies the algorithm. The negative phase of ionospheric storm is found using the reconstructed results of new algorithm. However, there are some grammar problems in the manuscript shown as follows:

1) In the Line 35 and 36, ionospheric vertical total electron content (VTEC) is computed should be revised as “ ionospheric vertical total electron content (VTEC) values are computed”. “Although the VTEC” should be revised as “ Although the VTEC image”.

2) in the Line 42, “has” should be revised as “have”.

3) in the Line 51, “the algorithm” should be revised as “ the algorithms”.

Reviewer 2 Report

The author proposes an algebraic reconstruction technique (ART) based on adaptive horizontal constraint and empirical orthogonal function (ARTHCEOF) to solve the tomographic inversion problem of the ionosphere. This work is an enjoyable read, also because it covers very interesting topics from a scientific point of view. I believe that after making the minor corrections the article can be considered for publication.

Line 68: Please explain the “dl” in Formula 1.

Line 80: Why NeQick model was chosen as the initial values?

Line 134: In figure2, only one ionsonde stations was chosen. What is the range of ionospheric measurements for this monitoring station.

Line 180: In Tabel 1, ARTHCEOF method only needs to iterate ten times, why does the iteration efficiency significantly improve compared to other methods.

Line 188: Please explain the “m” in Formula 8.

Line 226-227: In figure 10, why is the difference greater at high latitudes compared to low latitudes. Why it is not oceanic regions or areas without monitoring stations.

Reviewer 3 Report

The authors provided an improved constrained ART algorithm for ionospheric tomography. The superior tomographic performances were also validated by the ionosonde profiles and CODE TEC during different geomagnetic storms. It is recommended to be published in Remote Sensing after addressing some minor problems.

1. In Figure 9, the legends are all provided by solid lines. What do dash lines stand for? Please check.

2. Please add units of height, latitude, and longitude.

3. Figures 10 and 13, the captions of which read "The unit of TEC is 1016el/m2", while the units of the color bars are "TECU", Please unified.
